# Human muscle-derived CLEC14A-positive cells regenerate muscle independent of *PAX7*

Andreas Marg [1], Helena Escobar [1,2], Nikos Karaiskos[2,3], Stefanie A. Grunwald [1], Eric Metzler [1], Janine Kieshauer [1,2], Sascha Sauer[3,4], Diana Pasemann[5], Edoardo Malfatti[6,7], Dominique Mompoint[6], Susanna Quijano-Roy[6,7], Anastasiya Boltengagen [3], Joanna Schneider [1,4], Markus Schülke [8], Séverine Kunz [2,9], Robert Carlier [6,7], Carmen Birchmeier[2], Helge Amthor[6,7], Andreas Spuler[10], Christine Kocks [2,3], Nikolaus Rajewsky [2,3] & Simone Spuler [1,2,4]*

Skeletal muscle stem cells, called satellite cells and defined by the transcription factor PAX7, are responsible for postnatal muscle growth, homeostasis and regeneration. Attempts to utilize the regenerative potential of muscle stem cells for therapeutic purposes so far failed. We previously established the existence of human PAX7-positive cell colonies with high regenerative potential. We now identified PAX7-negative human muscle-derived cell colonies also positive for the myogenic markers desmin and MYF5. These include cells from a patient with a homozygous *PAX7* c.86-1G > A mutation (PAX7null). Single cell and bulk transcriptome analysis show high intra- and inter-donor heterogeneity and reveal the endothelial cell marker *CLEC14A* to be highly expressed in PAX7null cells. All PAX7-negative cell populations, including PAX7null, form myofibers after transplantation into mice, and regenerate muscle after reinjury. Transplanted PAX7neg cells repopulate the satellite cell niche where they re-express PAX7, or, strikingly, CLEC14A. In conclusion, transplanted human cells do not depend on PAX7 for muscle regeneration.

[1] Muscle Research Unit, Experimental and Clinical Research Center, a joint cooperation of Charité, Universitätsmedizin Berlin and the Max Delbrück Center for Molecular Medicine in the Helmholtz Association, Berlin, Germany. [2] Max Delbrück Center for Molecular Medicine in the Helmholtz Association, Berlin, Germany. [3] Berlin Institute of Medical Systems Biology (BIMSB) at the Max Delbrück Center for Molecular Medicine in the Helmholtz Association, Berlin, Germany. [4] Berlin Institute of Health, Berlin, Germany. [5] Department of Nuclear Medicine, Charité Universitätsmedizin Berlin, Berlin, Germany. [6] INSERM U1179, Université de Versailles Saint-Quentin-en-Yvelines, Versailles, France. [7] Hôpital Universitaire Raymond Poincare, Garches, France. [8] Department of Neuropediatrics, Charité Universitätsmedizin Berlin, Berlin, Germany. [9] Electron Microscopy Core Facility, Max Delbrück Center for Molecular Medicine, Berlin, Germany. [10] Department of Neurosurgery, HELIOS Klinikum Berlin-Buch, Berlin, Germany. *email: simone.spuler@charite.de

D espite our knowledge about the existence of a highly effective muscle regeneration program, muscle wasting disorders cannot yet be treated. Satellite cells are the tissue-specific stem cells of skeletal muscle[1]. Following muscle injury, satellite cells proliferate, differentiate, and fuse with existing muscle fibers to form new muscle tissue. Satellite cells also self-renew, thus securing their maintenance. The paired homeobox transcription factor Pax7 characterizes myogenic precursor cells, quiescent satellite cells and proliferating early myoblasts. Based on experiments in mice and mouse mutants, various functions of Pax7 were proposed and controversially discussed, for instance, a role in the specification or the postnatal maintenance of satellite cells[2–4]. Furthermore, conflicting reports on the function of Pax7 in muscle regeneration exist[5–9]. In addition, satellite cells expressing high levels of *Pax7* were reported to possess higher self-renewal capacity than Pax7-low cells[10]. *MYF5* is another transcription factor expressed in quiescent satellite cells. Myf5 may support myogenic commitment of satellite cells[11]. Attempts to utilize the regenerative potential of muscle stem cells for therapeutic purposes so far failed. Reasons are the low number of satellite cells, 3–6% of all myonuclei, difficulties to expand them while at the same time satellite cells fuse or go into senescence, the lack of migration from the injection site in allogeneic settings[12], and the lack of genetically corrected autologous cells in muscular dystrophies. The CRISPR/Cas9 technology may now allow for precise gene editing in primary cells. Finally, it is not clear which molecular markers define the cell populations with high myogenic potential. CD133 cells, PW1 cells and mesenchymal stem cells have all been proposed to have myogenic potential, but at least in mice there is no muscle regeneration without Pax7-positive satellite cells[6–8].

Muscle cells derived from induced pluripotent stem cells are also an option for therapeutic applications[13–15], but translation into clinics might be an only distant aim.

We aimed to evaluate the potential of primary human satellite cells and to identify subpopulations suitable for muscle regeneration. Previously, we established a method to expand human skeletal muscle-derived cells. These cells are grown out from small human muscle fiber fragments (HMFF). They are transplantable, and they contribute to muscle regeneration[16]. Here, we further characterize such cells and identified a new PAX7-negative myogenic cell population, characterized by CLEC14. Regeneration efficiency of myogenic desmin-positive cell populations did not depend on the expression level of PAX7.

## Results
### Characterization of human PAX7-positive, PAX7-negative, and PAX-null myogenic cell populations. Pure myogenic cell populations ($n = 103$) were isolated from human muscle biopsy specimens obtained from 25 different donors using manual dissection followed by hypothermic treatment (Fig. 1a; Supplementary Tables 1 and 2). All cell populations were 100% desmin-positive, but the percentage of nuclei that stained for PAX7 ranged from 0 to 98%. PAX7-positive (PAX7pos) cell colonies were defined as 15 to 98% PAX7-positive nuclei; PAX7-negative (PAX7neg) cell colonies had either 0% or 1% PAX7-positive nuclei. MYF5 was present in similar proportions of cells, i.e., was expressed independently of PAX7 (Fig. 1). We asked whether PAX7neg cell populations differed from PAX7pos cells in regard to their molecular profile and regenerative capacity. In addition, we included in our investigation cell colonies with a *PAX7*-null mutation. This patient carried a homozygous splice acceptor site mutation in *PAX7* c.86-1G > A, r.684_919del (NM_002584.2), which resulted in an exclusion of exon 2 and a premature stop codon in exon 3 (Supplementary Fig. 1, Supplementary Table 3

and 4)[17]. Other less likely pathogenic variants in the autozygous regions are depicted in Supplementary Table 3 and were determined by whole-exome sequencing. We did not find any de novo variant in exome of the index patient.

The 5-year-old girl presented with a congenital myopathy, rigid spine, and respiratory insufficiency. A detailed clinical description is provided (Material and Methods, Supplementary Fig. 1, Supplementary Video). The muscle biopsy specimen, obtained from M. rectus femoris, had no overt pathological alterations (Supplementary Fig. 1), but lacked PAX7-positive cells in tissue sections. Nevertheless, we identified cell colonies that grew out of the muscle fibers in our HMFF-based culture system. These cell colonies were desmin-positive, but did not express PAX7 mRNA or protein (PAX7null) (Fig. 1b–d).

We compared mRNA expression levels of various myogenic genes by qPCR in PAX7neg, PAX7null, and PAX7pos cells. In PAX7null cells, *PAX3*-mRNA was ~10-fold upregulated. PAX genes are crucial determinants of muscle development and PAX3 is the only other member of the gene family co-expressed with PAX7 in muscle precursors. *MYF5*, a key transcription factor for the myogenic lineage, was similarly expressed on mRNA and protein levels in all colonies, indicating that *MYF5* is expressed independently of *PAX7*. NCAM1 and MYOD1 mark satellite cells and myoblasts; both markers were strongly reduced in PAX7null cells (Fig. 1c). We also measured key markers of interstitial mesenchymal cell populations associated with some myogenic potential like fibroadipogenic cells, Osr1-positive, PW1/Peg3-positive cells, or mesangioblasts, respectively. (Fig. 1c; Supplementary Fig. 2a; Supplementary Table 6). PAX7null, but not PAX7neg or PAX7pos cells, expressed high levels of *PDGFRa* and *OSR1*, indicating that they potentially have fibroadipogenic character, but expression levels of CD29/ITGB1 were comparable. PEG3/PW1 also was low in PAX7null cells.

### Single cell and bulk RNA-Seq of human myogenic cell populations separated based on PAX7 expression. To gain further insight into the character of the various cell types, we employed single-cell sequencing (SCS) of a total of 66,000 single cells from nine colonies. These included PAX7pos and PAX7neg colonies from different donors (Donor A, C, G, H) as well as biological replicates. (Fig. 2; Supplementary Figs. 4–7; Supplementary Table 7). We sequenced an average of 5080 cells per population with a mean depth of 4753 reads per cell, and identified on average 1307 genes and 2305 transcripts (measured by UMIs) per cell. In parallel, bulk-mRNA-sequencing was performed (Supplementary Figs. 2c and 10d; Supplementary Table 8; Supplementary Data File 1). We created an interactive online tool to visualize expressed genes of interest in all cell populations https://shiny.mdc-berlin.de/hummus_sc_XkZL9gHZE2UBwjGb/).

We found that SCS and bulk RNA-Seq highly reproduced qPCR data in regard to the expression of myogenic genes like *PAX7, MYF5, MYOD1, NCAM1* (Figs. 1 and 2; Supplementary Figs. 2c, 4–7). tSNE plot analysis showed that cells from all analyzed colonies separated into different clusters. One of these clusters corresponded to proliferating cells (*MKI67, PCNA*; Fig. 2d; Supplementary Figs. 4–7), but others were identified independently of cell cycle signatures. PAX7pos colonies from all donors contained detectable amounts of PAX7 transcripts and were clearly distinct from PAX7neg colonies of the same donor; they also clustered differently than the cells of the PAX7null patient (Fig. 2). When all PAX7pos (or PAX7neg) colonies from different donors were directly compared, cell clustering was based on the donor (Supplementary Fig. 7a, b). We found highly expressed genes with no previous association to myogenic progenitor cells, such as *TFPI2, CLDN11, CLEC14A, COL6A3,*

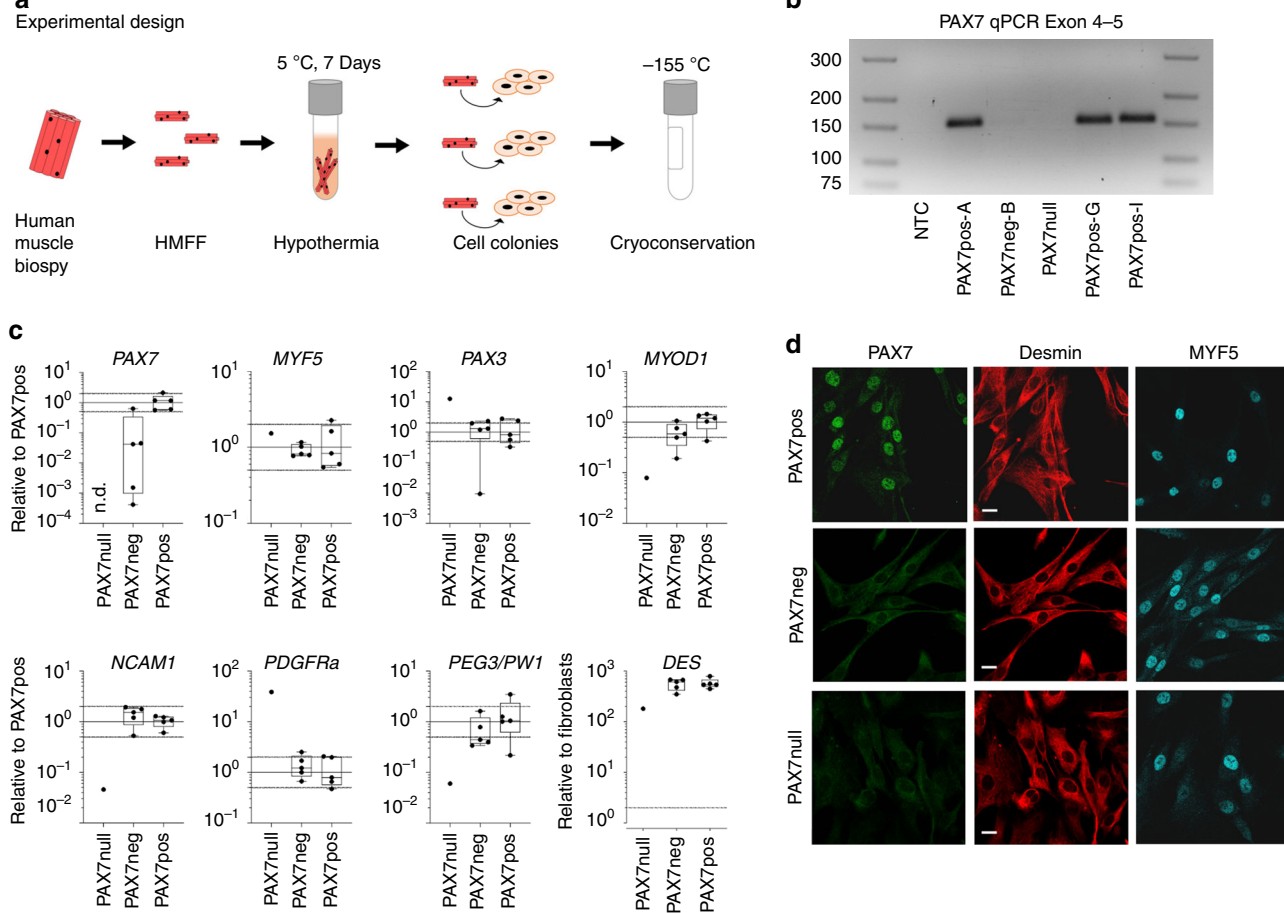

**Fig. 1 Characterization of human desmin-positive, PAX7- negative cell populations. a** Experimental design. Cell colonies grow out of human muscle fiber fragments (HMFF) within 3 weeks after hypothermic treatment. **b** Absence of *PAX7* transcripts in PAX7null cells. The *PAX7* c.86-1G > A mutation in PAX7null cells leads to deletion of exon 2 and a premature stop codon in exon 3. The PCR primers shown here recognize exons 4 and 5. PAX7neg-B cells are derived from donors with intact Pax7 gene and also do not express *PAX7*. **c** Expression of markers of myogenic and interstitial cells assessed by qPCR in PAX7null, PAX7neg, and PAX7pos cell colonies. All data were normalized to two reference genes and relativized to the mean of PAX7pos cells. The dotted line at twofold and 0.5-fold represents our threshold for differential expression. **d** Myogenic cells derived from HMFFs were stained for PAX7, desmin, and MYF5. PAX7-negative populations (PAX7null, PAX7neg) show no specific signal for PAX7 in immunofluorescence stainings. Cell lines are described in Supplementary Table 2. Scale bars: 20 μm.

and *FHL1*. Among these genes, CLEC14A was most abundantly expressed in PAX7null cells as determined in bulk RNA-Seq, qPCR, and by immunofluorescence (Supplementary Figs. 2c, 4a).

**Regenerative capacity of myogenic cell populations separated based on PAX7-expression.** After having established the molecular signatures of PAX7null, PAX7neg, and PAX7pos cells we evaluated their myogenic potential. We determined fusion indices in vitro and performed transplantation experiments in vivo. Fusion indices did not differ between PAX7neg, PAX7pos, and PAX7null cells with some variability within each group and ranged between 20 and 73% without serum depletion in either population (Fig. 3a, Supplementary Fig. 3, Supplementary Table 2). In vivo, we used a xenograft model. We injected $1 \times 10^5$ cells into the irradiated anterior tibial muscle (TA) of immuno-deficient NOG mice. Three weeks after transplantation, human muscle fibers had generated from PAX7pos, PAX7neg, and PAX7null populations as determined by human-specific anti-spectrin and anti-lamin A/C antibodies (Fig. 3b). Quantification of these experiments revealed that the number of human fibers as well as the fiber diameters was similar in transplants derived from either cell population (Fig. 3c). More strikingly, we found that

PAX7pos and PAX7neg cell colonies were able to repopulate the satellite cell niche between sarcolemma and basal lamina of muscle fibers, where they reexpressed PAX7 as assessed by immunohistological staining of laminin, human laminA/C and PAX7 (Fig. 3d). As expected, human muscle fibers derived from PAX7null cells did not express PAX7. It therefore came as a surprise that the satellite cell niche in human fibers derived from PAX7null cells was populated with CLEC14A-positive cells (staining with antibodies against laminin, human-specific lamin and human-specific CLEC14A) (Fig. 3d). We also performed reinjury experiments of irradiated and transplanted NOG mouse TA muscle using the myotoxic agent cardiotoxin (CTX). Injury of the irradiated muscles caused a severe atrophy of the TA muscles, but human muscle fibers and PAX7-positive cells in a satellite cell position were also detectable after injury. Remarkably, regeneration took also place after transplantation of PAX7null cells, and CLEC14A-positive cells were again observed in the satellite cell niche after reinjury (Supplementary Fig. 3). These results clearly show that human cells that are PAX7neg or PAX7null can generate new muscle fibers. CLEC14A-positive cells, located in a satellite cell position and generated in the absence of

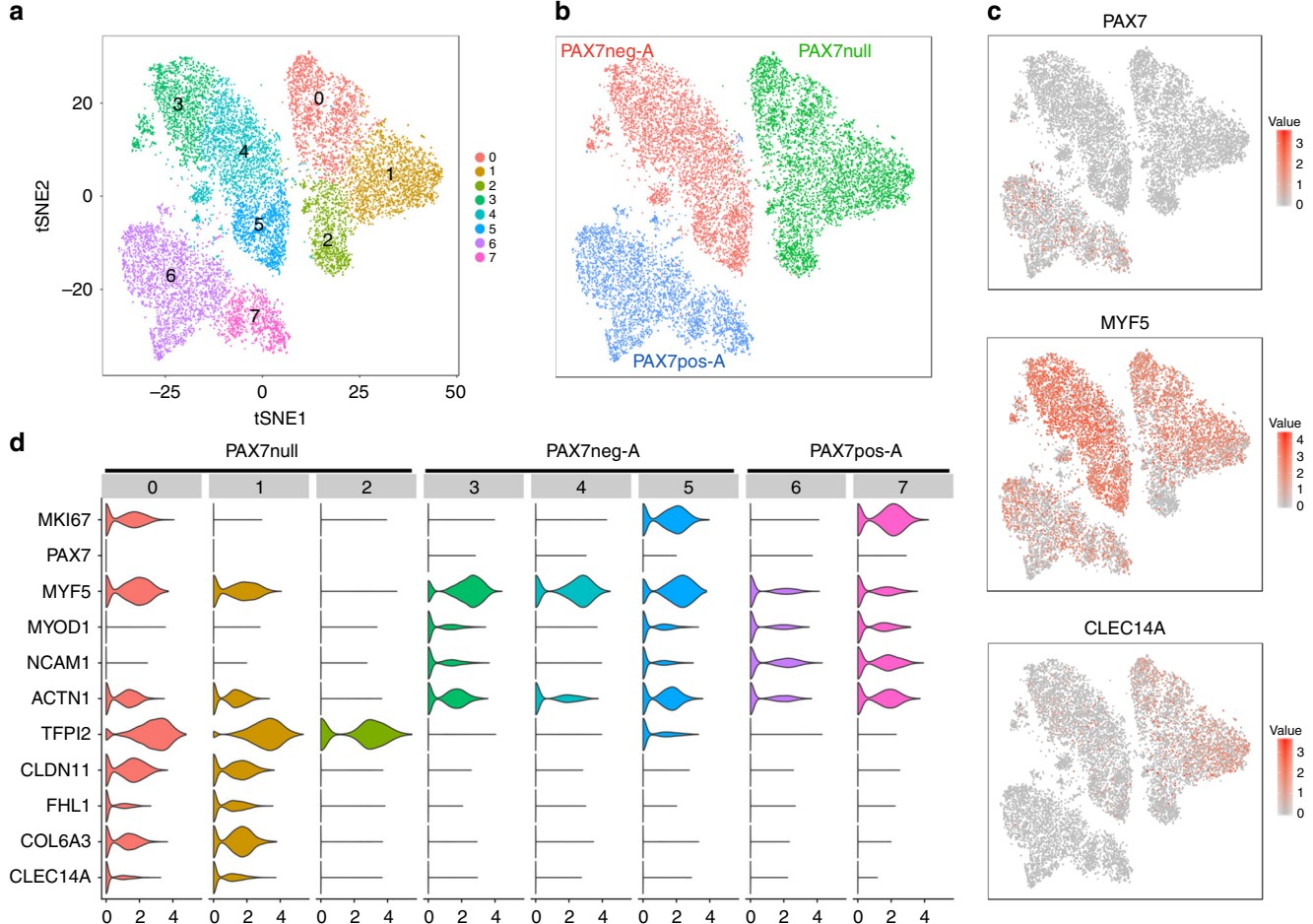

**Fig. 2 Single-cell transcriptomics reveals different gene expression pattern in PAX7pos, PAX7neg, and PAX7null myogenic cells. a–c** Two-dimensional tSNE representation of global gene expression relationships among individual myogenic cells from a PAX7-deficient donor (PAX7null) and from donor A (PAX7neg-A, PAX7pos-A). Each dot represents a cell. **a** tSNE plot colored by cell cluster. **b** tSNE plot colored by cell population. **c** Normalized gene expression levels of selected marker genes (red: high, gray: low). **d** Violin plots indicating gene expression levels and distributions of selected genes within cell clusters corresponding to panel **a**. Clusters 0, 5, and 7 correspond to proliferating cells (*MKI67*). PAX7null cells express strongly elevated levels of *CLDN11, FHL1, COL6A3,* and *CLEC14A* in clusters 0 and 1, and *TFPI2* in clusters 0, 1, and 2.

PAX7, appear to represent an additional stem cell pool for regeneration.

**Characterization of CLEC14A-positive cells and their regenerative capacity**. We characterized CLEC14A-positive cells in greater depth. PAX7null cells were strongly positive for CLEC14A (Fig. 4a). We then defined the abundance of CLEC14A-positive cells in the spectrum of HMFF-derived, myogenic 100% desmin-positive cell colonies from donors with functional *PAX7* genes (56 cell colonies, 24 donors,) (Supplementary Table 1) and found that ~20% of these cell colonies contain CLEC14A-positive cells (Fig. 4b). The distribution of PAX7pos and CLEC14A-positive cells in all 56 colonies indicated that PAX7 and CLEC14A expression was mutually exclusive (Fig. 4b). We confirmed this by double-immunofluorescence, which directly demonstrated that PAX7pos cells are negative for CLEC14A and vice versa (Fig. 4b). When CLEC14A-positive cells were sorted by fluorescence-activated cell sorting (FACS) and transplanted into NOG mice, human muscle fibers formed just as efficiently as when transplanting PAX7neg cell populations (compare Fig. 3b, c with Fig. 4d; Supplementary Fig. 8). Finally, we reexpressed PAX7 in PAX7null cells using a lentiviral *PAX7*-GFP expression vector and isolated transduced cells by FACS-sorting (Supplementary

Fig. 9). Induction of PAX7 drastically changed the molecular profile of PAX7null cells, inducing *MYOD1* and *NCAM1*, and downregulating CLEC14A (Fig. 4c). *MYF5* remained unaffected.

CLEC14A so far has only been described as an endothelial cell marker. We therefore performed molecular profiling and functional angiogenesis assays to elucidate possible endothelial characteristics of the myogenic, CLEC14A-positive cells. PAX7-null cells do not carry typical endothelial cell markers such as PECAM1 or ERG1 (Supplementary Fig. 10a). Their transcriptome clearly separates from HUVEC cells in principal component analyses (Supplementary Fig. 10d). Functionally, formation of tubes is observed with PAX7null as well as with PAX7neg or PAX7pos cells (Supplementary Fig. 10b). PAX7null cells depicted an unusual number of caveolae as would be expected in endothelial cells (Supplementary Fig. 10c).

At last, we returned to human muscle sections of the PAX7null patient to evaluate whether CLEC14A-positive cells are residents of the human satellite cell niche. Indeed, we found that the satellite cell niche in PAX7null muscle harbors resident cells that are positive for CLEC14A, syndecan-4 and PAX3 (Fig. 4e). In muscle sections from donors with wildtype PAX7, CLEC14A in satellite cells was more difficult to detect. The expression level in the presence of PAX7 is very low (Fig. 4b) and in

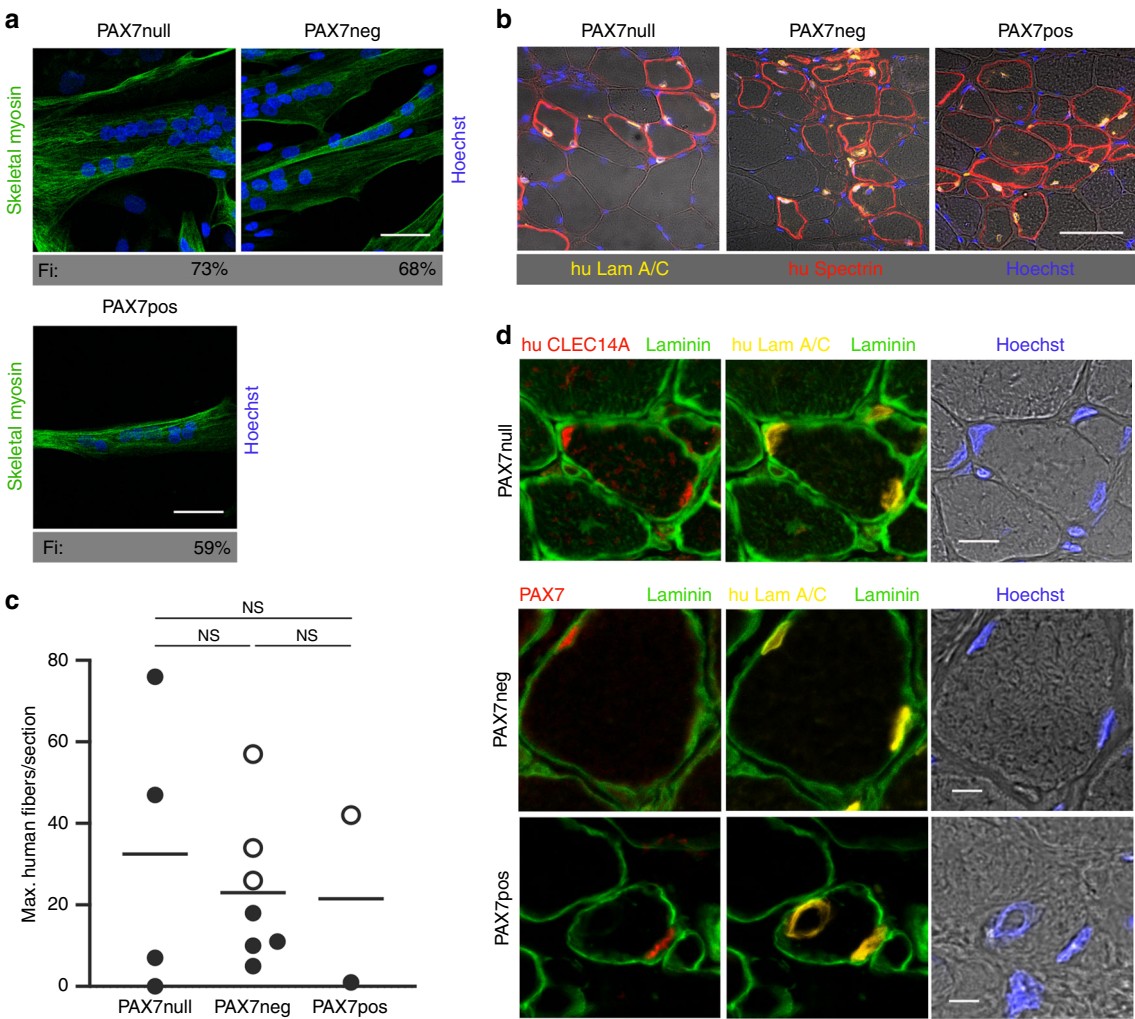

**Fig. 3 PAX7-negative cell populations can generate muscle. a** PAX7null, PAX7neg, and PAX7pos cells, used in transplantation experiments shown in **b**, indicate the same differentiation potential (skeletal myosin staining, Fi: Fusion index, Scale bars: 50 μm). **b** TA muscle sections from NOG mice demonstrate that 3 weeks after transplantation the PAX7null, PAX7neg, and PAX7pos human cell types contributed to muscle regeneration. Human anti-lamin A/C antibody (yellow) detects only human nuclei and anti-human spectrin antibody is specific for fibers of human origin (red). Nuclei in blue (Hoechst). Scale bar: 50 μm. **c** Quantification of experiments shown in **b**. Each dot represents one mouse grafted with the cells indicated below ($n = 4$ for PAX7null and $n = 7$ for PAX7neg). Shown are the values for the section with the highest number of human fibers for each mouse and the means. Only human muscle fibers with a diameter >10 μm were counted. Mice with >20 human fibers/section were additionally analyzed for PAX7 expression. The presence of PAX7-positive satellite cells is indicated by an open circle. The statistical differences between the groups were analyzed by two-tailed $t$ tests, $P > 0.05 =$ NS (not significant). **d** Repopulation of satellite cell niche after transplantation: In PAX7null transplants, the satellite cell niche is populated by CLEC14A-positive cells (red, upper row). After transplantation of PAX7neg- and PAX7pos cells, the satellite cell niche is populated by PAX7-positive cells (red, middle, and bottom row). Yellow, human lamin A/C; green, laminin; blue Hoechst. Scale bars: 10 μm (upper row) and 5 μm.

immunofluorescent stainings cannot compete with the strong surrounding signals from capillary endothelial cells. However, single molecule FISH (smFISH) generated the appropriate sensitivity to demonstrate that *CLEC14A* indeed is expressed in human satellite cells (Fig. 4f). Therefore, the presence of myogenic, desmin-positive, and CLEC14A-positive cell populations derived from HMFFs as well as mRNA expression in satellite cells in situ provide two lines of evidence that CLEC14A-positive cells belong to the normal spectrum of human muscle cells with regenerative potential in satellite cell position.

## Discussion

We report a human myogenic muscle stem cell, located within the satellite cell niche, that is defined by *CLEC14A* and *MYF5*. C-type lectin family 14 member A (CLEC14A) is a type 1

transmembrane protein specifically expressed and so far reported in endothelial cells[18]. MYF5 is a well-known transcription factor of the myogenic lineage downstream of PAX7. The novel cell type can well be identified in the patient carrying the PAX7null mutation, although it is not confined to this rare pathological condition. CLEC14A-positive myogenic cells also exist in individuals with functional *PAX7* genes. *CLEC14A* and *PAX7* expression is mutually exclusive. It will be interesting to further elucidate the mechanism on how the gene expression of *PAX7*, *MYOD1*, *MYF5*, *NCAM1*, and *CLEC14A* are regulated and intercalated with each other in human muscle stem cells. *MYF5* expression appears to be regulated independently from *PAX7*, *MYOD1*, and *CLEC14A*.

The patient with the PAX7null mutation displays features that resemble the reported phenotype of *Pax7*-deficient mice[2]. As seen in whole-body MRI, the patient had small but normal muscle

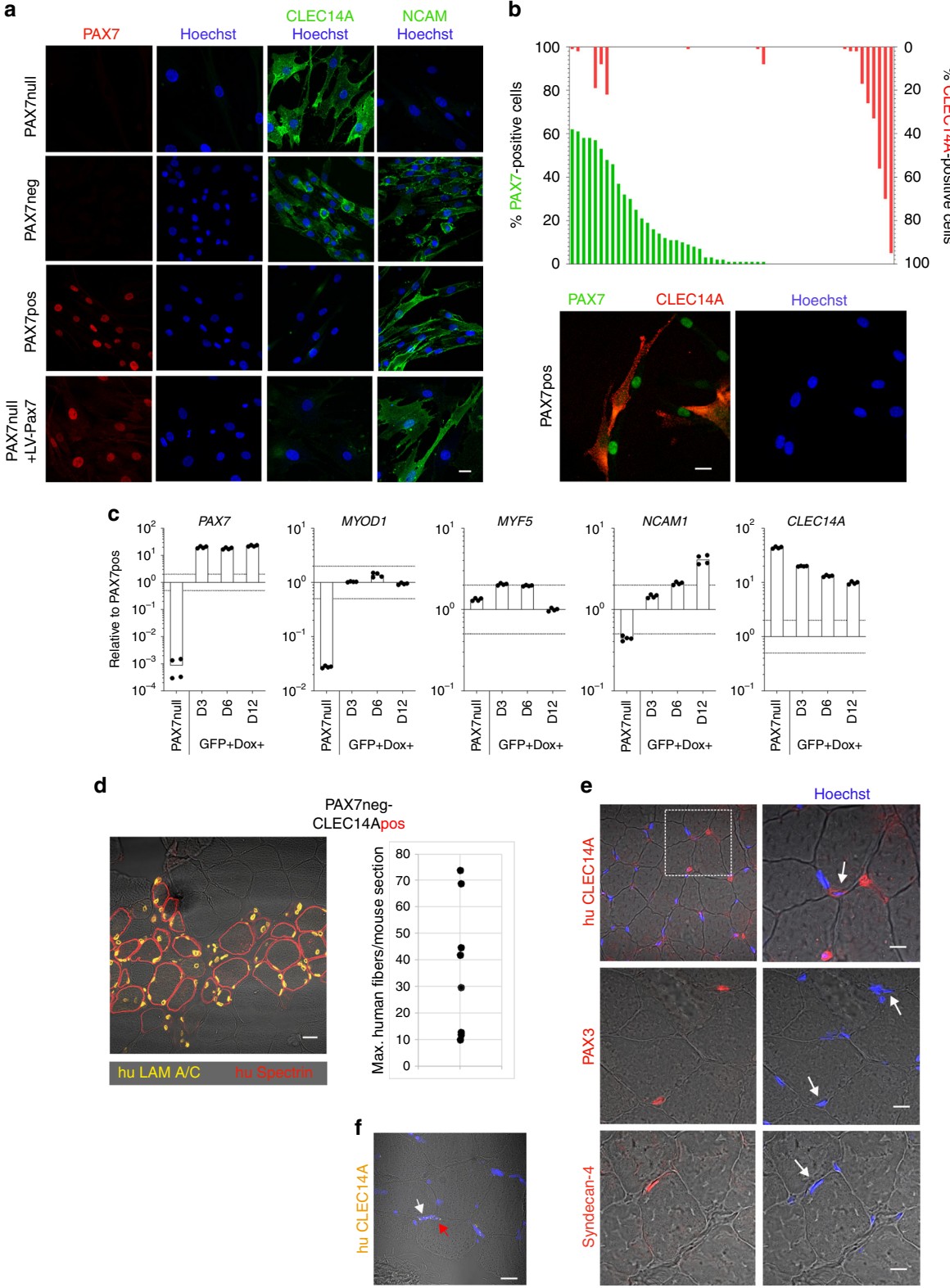

patterning, indicating that all muscles had formed in development. Pax genes are crucial for organogenesis, and Pax3 is the only other member of the gene family co-expressed with Pax7 in myogenic cells. *PAX3* mutations in Klein–Waardenburg syndrome lead to distal muscle defects[19], whereas a *PAX7*null mutation, as shown here, predominantly affects axial muscle raising the question as to whether these genes are differentially expressed in satellite cells of different muscles. PAX3-positive satellite cells define a subpopulation of muscle satellite cells that are particularly resistant to stress[20,21]. PAX7null cells also express PAX3. Head muscle formation is controlled by a distinct genetic program not involving Pax3/7 in mice. The presence of the severe congenital ptosis in the PAX7null patient indicates that the role of *PAX7* in head muscles needs further investigation.

**Fig. 4 CLEC14A is a marker for a human myogenic cell in satellite cell position. a** CLEC14A- staining of PAX7null, PAX7neg, PAX7pos cells. PAX7null cells are strongly positive for CLEC14A. Some PAX7neg cell populations from dornors with intact PAX7 gene are also CLEC14A-positive. Reconstitution of PAX7null cells with a lentiviral PAX7 vector changed the expression profile of CLEC14A and NCAM (bottom line). Red, PAX7; green, CLEC14A; blue, Hoechst. Bar 20 µm. **b** PAX7 and CLEC14A expression in PAX7neg cell colonies were mutual exclusive. Fifty-six cell populations from donors with intact PAX7 genes were analyzed for expression of PAX7 (green) and CLEC14A (red). Cell colonies are either positive for PAX7 or for CLEC14A. Some colonies contained both, cells that were either PAX7 or CLEC14A-positive. Staining for PAX7 and CLEC14A in "mixed" cell populations depicted mutually exclusive expression. Scale bar: 20 µm. **c** Gene expression in PAX7null cells selected by FACS-sorting (see also Supplementary Fig. 9) after lentiviral PAX7 transduction determined by qPCR 3, 6, and 12 days after DOX induction. All data were normalized to two reference genes and relativized to the mean of PAX7pos cell colonies. The dotted line at twofold and 0.5-fold represents the threshold for differential expression. **d** TA muscle section from a NOG mouse 3 weeks after transplantation of CLEC14Apos cells shows newly built human muscle fibers. Quantification reveals similar results as in Fig. 3b. Quantification as in 3c, but without PAX7 analysis. Each dot represents one mouse grafted with PAX7neg-CLEC14Apos cells ($n = 8$). Scale bar: 20 µm. **e** Staining for huCLEC14A, PAX3 and Syndecan-4 indicates cells in satellite cell position (arrows) in the muscle tissue of the PAX7null patient. Nuclei are stained with Hoechst (blue). About 8% of nuclei were positive for CLEC14A and localized in a satellite cell position (133 nuclei analyzed). Scale bars: 10 µm. **f** SmFISH for huCLEC14A in a frozen muscle section of a normal human donor. The red arrow indicates a nucleus in satellite cell position and the white one a endothelial cell nucleus. Scale bar: 20 µm.

Apart from CLEC14A, a number of genes were highly expressed in PAX7null cells that were not previously associated to myogenic progenitor cells, such as *TFPI2, CLDN11, COL6A3,* and *FHL1*. These genes are interesting for several reasons: First, these gene expression signatures might provide further insight about the unusual myogenic PAX7null cells. Tissue factor pathway inhibitor-2 (*TFPI2*) is a serine protease inhibitor expressed in vascular endothelium, and is required for cardiogenesis in zebrafish[22,23]. *CLDN11* is a marker of endothelial cell tight junctions[24,25]. Thus, despite the fact that these cells express myogenic markers like *PAX3* and *MYF5*, they also express genes typical for endothelial (*CLEC14A, CLDN11*) or fibroadipogenic cells (*PDGFRa* and *OSR1*). Second, from a translational point of view, it is interesting that mutations in *COL6A3* and *FHL1*, both highly expressed in PAX7null cells, cause phenotypically very distinct muscle diseases (Ulrich congenital muscular dystrophy and reducing body myopathy, respectively), both characterized by rigidity of the spine and respiratory failure[26–29], which are also prominent findings in the PAX7null patient.

Our findings do not allow conclusions as to whether myogenic cells from the PAX7null patient originate from embryonic endothelial cells, but a lineage relation between these cells types exists in development, and even in the adult endothelial cells[30–32]. Lineage-tracing experiments are not possible within our setup. Key markers of many cell types that have previously been shown to contribute to muscle regeneration like *PDGFRa, PDGFRb, PEG3/PW1, OSR1, ITGB1/CD29, TCF4, TP53,* and *PROM1/ CD133* have been analyzed[33–40]. CLEC14A-positive myogenic cells display features of FAPs because they express *PDGFRa* and *OSR1*, but ITGB1/CD29 does not differ from PAX7pos cells. Also, they do not differentially express PW1 or PDGFRb. They also do not share the features of mesangioblasts (CD133) or endothelial cells. Furthermore, in contrast to FAPs or mesangioblasts, CLEC14A-positive myogenic cells inhabit the satellite cell niche. Apart from CLEC14A, other endothelial cell markers such as ERG1, PECAM, or VCAM were not expressed in CLEC14A-positive human myogenic cells[41,42]. We conclude that there is an overlap between features of CLEC14A-positive myogenic cells and previously described cell populations, but CLEC14A apprears to define a so far non-described population in human muscle.

In summary, human muscle fiber regenerate after transplantation and after reinjury in the same abundance, independent of the expression of PAX7. PAX7null cells with myogenic potential express CLEC14A. CLEC14A-positive cells belong to the normal spectrum in humans and are not confined to the rare patient carrying the PAX7null mutation. They have myogenic and regenerative capacity and could become relevant in cell therapies involving muscle stem cells.

## Materials and methods

**Patients**. HMFFs were prepared from muscle biopsy specimens. Research use of the material was approved by the regulatory agencies (EA1/203/08, EA2/051/10, EA2/175/17, Charité Universitätsmedizin Berlin). Informed consent was obtained from the donors or the legal guardians.

**Case report of patient with a homozygous *PAX7* c.86-1G>A mutation (PAX7null)**. The patient was a 5-year-old girl born to fourth degree con-sanguineous parents (first cousins) of Kurdish origin, with no family history for neuromuscular disorders. Her two younger siblings were healthy. Her mother reported scarce fetal movements during pregnancy. She was born at term and suffered from an amniotic infection syndrome requiring antibiotic treatment and oxygen supply. She had congenital severe bilateral ptosis that required head extension for upward gaze. She was bottle fed owing to poor suction. She had normal spontaneous limb movements and started rolling over at 3–4 months of age. She could not lift her head at ventral position and failed to crawl. She achieved independent sitting at 5–6 months of age and standing position at 10 months. She walked independently at 14 months of age and was able to climb stairs from two years of age onwards. She fell frequently that was likely provoked by an impaired downside field of view as she compensated her ptosis by head extension. In addition, she developed increasing difficulties for stair climbing and required pulling at the ramp and pushing on her upper legs with her arms. However, she could climb walls independently as leisure activity. Walking became more difficult with walking perimeter below one kilometer because of rapid fatigability. She developed an increasing hyperlordosis. At the age of 5 years she developed a respiratory insufficiency requiring short-term invasive assisted ventilation during a respiratory infection. Nocturnal non-invasive-assisted ventilation was started to compensate sleep disordered breathing and nocturnal hypoventilation. She had normal cognitive and speech development and normal sensory abilities. Body length was at 50th percentile (P.) at birth, 75th P. at 4 years and dropped to 25th P. near 6 years of age. Body weight fall from 15th P. at birth to under 3rd P. at near 6 years of age.

Clinical examination at near 6 years of age revealed global muscle hypotrophy, rigid spine with hyperlordosis and mild scoliosis, scapular winging, and flattened thorax (Supplementary Fig. 1, Supplementary Video). She had mild dysmorphic facial features associated with severe bilateral ptosis, decreased orbicular oculi strength without facial weakness. External eye movements were only limited for extreme upwards gaze. Proximal lower limb strength was reduced, notably for thigh flexion and adduction. Upper arm movements above 90° were nearly impossible. She had prominent axial muscle weakness with profound neck flexor and extensor weakness (head extension and flexion were impossible against gravity). In contrast, sideward head movements were nearly normal. No muscle weakness was noticed in distal segments.

Forced vital capacity (FVC) of 0.33 l was measured in sitting position, corresponding to 35% predicted FVC (FVC%p). Ultrasonography revealed weak diaphragmatic motion, diaphragm atrophy as well as lack of inspiratory diaphragm thickening (Supplementary Video). Sensitive and motor nerve conduction studies and repetitive stimulation were performed on upper limbs and did not reveal signs of neuropathy or abnormal neuromuscular transmission (data not shown).

Whole-body muscle MRI (Supplementary Fig. 1, Supplementary Video), compared with an age-matched child with no neuromuscular disorder, revealed normal masticatory muscles and tongue. Occipital muscles were almost entirely replaced by fatty-fibrotic tissue, with only small muscle bands persisting. At neck level, levator scapulae muscles were well conserved, whereas dorsal neck, trapezius superior, and sternocleidomastoideus muscles as well as head flexors persisted only as residual bands. Shoulder girdle, upper limb, and hand muscles were normally developed with no pathological signature. At body wall level, extrinsic back muscles and intercostal muscles were extremely hypotrophic and largely immured

in fatty-fibrotic tissue. Abdominal wall muscles were hypotrophic. At pelvic girdle, gluteal muscles were hypotrophic at proximal level, whereas their distal portions were missing altogether. Psoas and iliac muscles persisted only as small bands. At thigh level, rectus femoris was well preserved, whereas other thigh muscles were hypotrophic and replaced by fatty tissue to various extents, with notably severe pathological changes noted for abductor femoris, the proximal part of vastus lateralis and distal part of vastus medialis muscles. Lower leg muscles and feet muscles were largely unaffected apart from some mild fatty infiltration of the soleus muscles. Biplanar imaging of the skeleton using EOS system identified no developmental skeletal abnormalities with notably normal skeletal elements of the thorax and spine (Fig. 4c). However, EOS confirmed the presence of scoliotic and hyperlordodic deformation of the spine together with a flattened thoracic cage.

Both parents were healthy and strictly normal at neurological examination. In particular they had normal muscle bulk and strength. Whole-body MRI failed to show any abnormality; the probands father featured a particularly well-developed muscle bulk (data not shown).

**Satellite cell cultures**. HMFFs were prepared according to Marg et al.[16] with following modifications. Hypothermic treatment was performed at 5 °C and the incubation time was restricted to 7days. Up to 24 cell colonies grow out from each muscle biopsy. Selected colonies were frozen at − 155 °C for long-time storage.

**Immunofluorescence LSM microscopy**. Cells were grown on μ-Slides (eight-well, ibidi) for 2 days, washed with phosphate-buffered saline (PBS) and fixed with 3.7% formaldehyde. After permeabilization with 0.2% Triton X-100 (except CLEC14A- and NCAM staining), the cells were incubated in 5% bovine serum albumin (BSA)/ PBS for 1 h. Primary antibodies were used as described in Supplementary Table 5 and incubated in 1% BSA/PBS for 3 h at room temperature (RT). After washing, samples were incubated with Alexa 488-, Alexa 568-, or Alexa 647-conjugated secondary antibodies (Thermo Fisher Scientific; each 1:500) and Hoechst 33258 (0.5 μg/ml, Sigma-Aldrich) for 1 h in PBS at room temperature. For Co-Staining: Cells were stained first with CLEC14A as described. Afterwards the cells were permeabilized with 0.2% Triton X-100 and the protocol for PAX7 was run overnight.

Muscle samples (M. rectus femoris from the PAX7null patient or M. tib. ant. from cell-transplanted NOG mice) were frozen in liquid nitrogen-cooled isopentane and 6 μm-thick sections were cut with a Leica cryostat (CM 3050 S). Fresh (PAX3 and PAX7) or frozen cryosections were fixed with 3.7% formaldehyde (CLEC14A, PAX3, PAX7, and Syndecan-4) or aceton. Primary antibody incubation was done overnight (CLEC14A, PAX3, PAX7 and Syndecan-4) or for 2 h (Supplementary Table 5). Samples were imaged with a Zeiss LSM 700 confocal microscope (Carl Zeiss MicroImaging GmbH) or an EVOS FL Cell Imaging System (Thermo Fisher Scientific). Images were composed and edited in ZEN 2.3 (Carl Zeiss Microscopy GmbH) and CorelDRAW 2017.

**Single-molecule fluorescence in situ hybridization**. SmFISH was performed with 10 μm- fresh-frozen human muscle sections according to the manufacturer's manual (ACD, 323100-USM/Rev Date: 02272019). To avoid over-digestion protease IV incubation time was reduced to 10 min. For CLEC14A mRNA- detection RNAscope Probe -Hs-CLEC14A-C2 (ACD, Catalog number: 510761-C2) was used.

**Differentiation assay**. Muscle cell differentiation was done without a specific differentiation medium. In all, 5000 cells per well were seeded on a 24-well plate in skeletal muscle cell growth medium with fetal calf serum (FCS, Provitro). After 9 days in a humidified atmosphere containing 5% $CO_2$ at 37 °C, cultures were fixed and stained with an antibody against skeletal myosin (Supplementary Table 5), nuclei were stained with Hoechst. The fusion index was determined by dividing the number of nuclei within myotubes by the total number of nuclei counted. For each sample at least 600 nuclei were counted.

**Human PAX7 protein expression in human cells**. The human *PAX7*-coding sequence was purchased from DNASU Plasmid Repository (Clone: HsCD00513858; NCBI Reference Sequence: NM_013945.2, bases 599–2155). The original TAA Stop-Codon was inserted via PCR at position 2152. The *PAX7*-coding sequence was inserted into the lentiviral backbone vector pINDUCER21 (ORF_EG) (gift from Stephen Elledge & Thomas Westbrook; Addgene plasmid #469489)[37] using the Gateway Cloning Technology (Life technologies). The Lentivirus was produced using the PLP1, PLP2, and VSV-G production plasmids (Thermo Fisher Scientific) in HEK293TN cells. In total, $4 \times 10^6$ PAX7null cells ($5.2 \times 10^3$/cm$^2$) were infected with a MOI of 7.8 and FACS-sorted (FACSAria Fusion, BD Biosciences) 4 days after infection using a GFP reporter. Exogenous PAX7 expression was induced 3 days after FACS-sort by 8 ng/ml doxycycline. Cells were collected 3, 6, and 12 days after induction. Uninfected PAX7null cells were cultured parallel to the lentivirus infection protocol and used as control.

**Whole-exome sequencing**. DNA was isolated from peripheral leukocytes. Whole-exome sequencing (WES) was done in the index patient and in both her consanguineous parents (WES trio). Exonic sequences and flanking intronic regions

were captured using the BGISEQ-500 whole-exome enrichment and sequencing protocol[43] yielding ≈ 75 Mio paired-end FASTQ reads. These were aligned to the human GRCh37.p11 genomic reference with BWA-MEM v0.7.1[44]. The mean coverage was 130–154×, > 97% of the RefSeq positions were covered > 10×, and > 93% > 20×. After fine-adjustment, the raw alignments were called for deviations from the reference in all coding exons and 50-bp flanking regions using GATK v3.8[45]. The resulting VCF file was analyzed with MutationTaster2 to assess potential pathogenicity of all variants[46]. Variants were filtered for different inheritance models. For the recessive inheritance model, we removed variants that occurred > 30× in homozygous state in the gnomAD database (homozygote frequency > 6.0E-05). In addition, owing to the consanguinity of the parents, we performed an autozygosity mapping with the HomozygosityMapper2012 software[47] using the VCF-files of the family members. Only variants were considered as pathogenic that were located in genomic regions that were homozygous in the index patient, but not in her parents. These genomic regions comprised a total of 227.8 Mbp. For the dominant inheritance model, we removed variants that occurred > 25 × in the heterozygous state in the gnomAD database (MAF > 1.0E-04). For the de novo model, we searched for variants present in the index patient, but absent from both parents.

Beyond that we investigated virtual panels of genes that are associated with diseases annotated with "rigid spine" (HPO:0003306) in the OMIM database. Additional panels comprised $n = 35$ genes for "Congenital myasthenic syndrome" and $n = 101$ genes for "Congenital myopathy". Potentially disease-causing variants were further assessed for their pathogenicity using the additional information provided by MutationTaster2 at http://www.mutationtaster.org (accessed March 2019) and visually inspected using the IGV software downloaded from http://www.broadinstitute.org/igv/.

**Virtual gene panels**. Panel 1: Muscle diseases annotated with "rigid spine" in OMIM (n=24): *ACTA1, ACVR1, BAG3, BIN1, COL6A1, COL6A2, COL6A3, DES, EMD, FHL1, KY, LMNA, MGME1, NEB, ORAI1, POMT1, POMT2, RYR1, SELENON, TNPO3, TOR1AIP1, TPM3, TRIP4, VAMP1*

Panel 2: Congenital myopathy, Version 1.129 (n=101), Genomics England PanelApp at https://panelapp.genomicsengland.co.uk/panels/225/: *ACTA1, ACTN2, AR, ATP2A1, BAG3, BIN1, CACNA1S, CASQ1, CAV3, CCDC78, CFL2, CHCHD10, CNTN1, COL12A1, COL6A1, COL6A2, COL6A3, COL9A3, CPT2, CRYAB, DES, DMPK, DMPK, DNAJB6, DNM2, DOK7, DYSF, ECEL1, EPG5, FAM111B, FKBP14, FLNC, GFER, GNE, HACD1, HNRNPA1, HRAS, HTRA2, ISCU, KBTBD13, KLHL40, KLHL41, KLHL9, KY, LAMP2, LDB3, LGI4, LMNA, LMOD3, MAP3K20, MATR3, MEGF10, MICU1, MTM1, MTMR14, MT-TL1, MYBPC1, MYBPC3, MYF6, MYH14, MYH2, MYH3, MYH7, MYH8, MYL1, MYMK, MYO18B, MYOT, MYPN, NEB, NEFL, ORAI1, PIEZO2, PNPLA2, PUS1, RBCK1, RYR1, SCN4A, SELENON, SLC25A4, SLC25A42, SPEG, SPTBN4, SRPK3, STAC3, STIM1, STIM2, TIA1, TNNC2, TNNI2, TNNT1, TNNT3, TPM2, TPM3, TRIP4, TTN, VCP, VMA21, VPS33B, YARS2, ZC4H2*

Panel 3: Congenital myasthenic syndrome, Version 1.47 (n=35), Genomics England PanelApp at https://panelapp.genomicsengland.co.uk/panels/232/: *AGRN, ALG14, ALG2, CACNA1A, CHAT, CHRNA1, CHRNB1, CHRND, CHRNE, CHRNG, COL13A1, COLQ, DOK7, DPAGT1, GFPT1, GMPPB, LAMA5, LAMB2, LRP4, MUSK, MYO9A, PLEC, PREPL, RAPSN, RYR1, SCN4A, SLC18A3, SLC25A1, SLC5A7, SNAP25, SYT15, SYT2, TOR1AIP1, UNC13A, VAMP1*

**Targeted sanger sequencing for PAX7**. Genomic DNA from patient PAX7null was isolated from blood with Qiagen FlexiGene DNA kit (Qiagen) and amplified using the Taq "all inclusive" PCR reaction kit (PeqLab). PAX7 primer sequences based on NG_023262.1 are listed in Supplementary Table 4. Sequencing at genomic DNA level was performed by Source Bioscience. CLC Genomics workbench (v9.5) was used for sequence alignments.

**RNA isolation, reverse transcription, and quantitative PCR**. Total RNA was isolated from human primary myoblasts and skeletal muscle tissue using NucleoSpin RNA/Protein Kit (Marcherey-Nagel) and treated with DNAse for genomic DNA removal. RNA quantity and purity were determined with a NanoDrop ND-1000 spectrophotometer (Thermo Scientific). RNA was reverse transcribed through the QuantiTect reverse transcription kit (Qiagen). All qPCR experiments were performed according to the MIQE guidelines[48] using KAPA SYBR FAST qPCR MasterMix Universal (PeqLab) in the Mx3000P instrument (Stratagene) (Primer sequences Supplementary Table 6). GAPDH and cyclophilin A were used as reference genes. GAPDH normalized data were shown and were equivalent to calculations using cyclophilin A. qPCR results were analyzed with MxPro software (v4.1), calculated in Excel, and plotted using GraphPad Prism (v8).

**Bulk RNA sequencing (RNA-seq)**. RNA Sequencing libraries were generated using the NEBNext rRNA Depletion Kit and NEBNext Ultra Directional RNA Library Prep Kit for Illumina (NEB). Libraries were sequenced using a MiSeq system (Illumina) in paired-end mode (read length 75nt). HUVEC read counts were obtained from GEO (SRR7549223, SRR7549223, SRR7549225). Pre-processed RNA-Seq data were analyzed using CLC Genomics Workbench (v. 9.5, Qiagen) with a workflow executing the following the steps: (1) quality read trimming

(2) quality control (3) RNA-Seq analysis by aligning reads to the human reference genome and transcriptome (GRCh38). All samples passed the RNA-seq quality check (Supplementary table 8). Total read counts were normalized by transcripts per million (TPM). CLUSTVis was used for heatmap and PCA generation. Read counts, normalized count data (Supplementary Data File 1), and raw fastq files are uploaded to GEO BioProject PRJNA481958.

**Single-cell sequencing: drop-seq procedure, single-cell library generation, and sequencing**. The Drop-seq procedure[49] allows cost-effective highly parallel sequencing of single cells and detects 3′ ends of messenger RNA and long-noncoding RNA molecules. Upon nanoliter droplet formation, individual cells are co-encapsulated with individual, uniquely barcoded beads, and become lysed completely. Released cellular RNA transcripts then hybridize via their polyA tails to polyd(T) primers that are attached to uniquely barcoded beads carrying unique molecular identifiers (UMIs). Nanoliter-droplets are collected and broken, and transcripts reverse transcribed into complementary DNA (cDNA), amplified by PCR and sequenced in bulk.

Monodisperse droplets of ~1 nl in size were prepared on a self-built Drop-seq set up following closely the instrument set up and library generation procedures invented by Macosko et al.[49] with minor modifications as previously described[45]. Frozen cells (~1 million) were thawed immediately before a Drop-seq run, and subsequently kept on ice or handled in the cold. Usually, a small number of cells was seeded into cell culture flasks and expanded for immunofluorescence staining to ensure cell characterization after thawing. The remaining cells were centrifuged at 150×*g* for 5 min, resuspended in phosphate-buffered saline pH7.2 supplemented with 0.01% bovine serum albumin fraction V (Biomol, 01400) and 1 U/µl RiboLock RNase inhibitor (Thermo Fisher Scientific, EO0381), centrifuged, resuspended again, strained through a 35 µm nylon cell strainer (Corning, 352235), counted and used for Drop-seq at a final input concentration of ~200 cells/µl. Cell viability was assessed by Trypan blue staining and was between 70 and 90% in all cases.

Individual cDNA libraries ranged from ~1.0 to 1.7 kb in average size, indicating good overall quality of RNA and cDNA molecules. Single-cell Drop-seq libraries at 1.8 pM (final insert size average ~700 bp) were sequenced in paired-end mode on Illumina Nextseq500 sequencers with 1% PhiX spike-in for run quality control using Illumina Nextseq500/550 High Output v2 kits (75 cycles). Read 1: 20 bp (bases 1–12 cell barcode, bases 13–20 UMI; Drop-seq custom primer 1 "Read1CustSeqB"), index read: 8 bp, read 2 (paired end): 64 bp.

**Computational methods in single-cell RNA-seq**. *Data processing, alignment, and gene quantification*: We chose read 1 to be 20-bp long, which sequences the cell barcode at positions 1–12 and the UMI at positions 13–20, while avoiding reading into the poly(A) tail. The remaining 64 sequencing cycles were used for read 2. Sequencing quality was assessed by FastQC 2(v.0.11.2), while special attention was paid to the base qualities of read 1 to assure accurate cell and UMI calling. We used the Drop-seq tools v. 1.12[45] to trim poly(A) stretches and potential SMART adapter contaminants from read 2, to add the cell and molecular barcodes to the sequences and to filter out barcodes with low quality bases. The reads were then aligned to the reference genome hg38 (Ensemble annotation 84). Typically, ~75–80% of the reads were found to uniquely map to the genome; multi-mapping reads were discarded. The Drop-seq toolkit was further used to add gene annotation tags to the aligned reads and to identify and correct bead synthesis errors, in particular base missing cases in the cell barcode. Cell numbers were estimated by plotting the cumulative fraction of reads per cell against the cell barcodes and calculating the knee point. The DigitalExpression tool[49] was used to obtain the digital gene expression matrix (DGE) for each sample.

*Cell filtering and data normalization*: For a first view of gene quantification and basic statistics we used the R package "dropbead" (https://github.com/rajewsky-lab/dropbead)[46]. We discarded cells expressing < 500 UMIs, or more cells in which the mitochondrial encoding RNA was > 35%. We normalized the UMI counts for every gene per cell by dividing its UMI count by the sum of total UMIs in that cell, and multiplying it by the number of UMIs that the deepest cell contained. Downstream analysis was performed in log space.

*Clustering, t-SNE representation and marker discovery*: We used Seurat[50] to identify highly variable genes, perform principal component analysis, identify the most important principal components and use them for clustering and t-SNE representation. Around ~1200 highly variable genes were identified and the first 20 principal components were used for the clustering and the tSNE representation. We used Seurat's function FindAllMarkers to find potential markers for each cluster in an unsupervised manner.

**FACS of CLEC14A-positive/negative cells**. Cultured human primary myoblasts were detached by incubating with 0.25% Trypsin-EDTA for 4 min at 37 °C. Cells were collected in Skeletal Muscle Cell Growth Medium (SMCGM, Provitro) and centrifuged for 5 min at 200×*g*. The resulting cell pellet was resuspended in ice-cold sterile-filtered 1% BSA in PBS (staining buffer) and centrifuged again at 200×*g* for 5 min at 4 °C. Cells were resuspended in staining buffer containing Anti-CLEC14A antibody (ThermoFisher #PA5-47677, conc. 0.2 µg/µl, dilution 1:50) and incubated for 20 min at 4 °C. Cells were washed twice with ice-cold PBS and incubated for 15 min at 4 °C in staining buffer containing Alexa Fluor 488-conjugated donkey

anti-sheep antibody (ThermoFisher #A-11015, conc. 2 µg/µl, dilution 1:500). Following incubation with the secondary antibody, cells were washed twice with ice-cold PBS, resuspended in 1 ml ice-cold staining buffer containing 100 µg/ml Primocin (InvivoGen #ant-pm-1) and transferred to FACS tubes with 40 µm cell strainer caps to remove cell clumps. CLEC14A-positive and negative cells were sorted using a BD FACSAria Fusion Cell Sorter. After sorting, cells were collected and cultured for 2 days in SMCGM containing 100 µg/ml Primocin and two more days in SMCGM only prior to transplantation.

**Animal experiments**. All animal experiments were performed under the license number G0035/14 (LAGeSo, Berlin, Germany). In all, 5- to 7-week old male NOD.Cg-*Prkdc^scid^Il2rg^tm1Sug^*/JicTac (NOG) or NOD.Cg-*Prkdc^scid^Il2rg^tm1Wjl^*/SzJ (NSG) mice were purchased from Taconic Biosciences or Charles River Laboratories, respectively, 1 week before each experiment. They were kept under daily monitoring at our specific pathogen-free animal facility with free access to food and water, provided with hiding place and nest material. Hygienic monitoring was done according to FELASA recommendations.

**Intramuscular cell transplantation**. Focal irradiation of the recipient hind limbs was performed 2–3 days prior to cell transplantation as described[16,51]. In brief, mice placed under 9 mg/ml ketamine–1.2 mg/ml xylazine anesthesia at a dose of 160 µl/20 g were treated with a single X-radiation dose of 16–18 Gy using a CyberKnife image-guided robotic radiosurgery system. The radiation protocol was calculated according to a computed tomography scan, with 0.75 mm slice thickness, and was devised to deliver the desired dose in the three-dimensional target area within ~2 min, sparing the adjacent body parts. Biafine lotion was applied on the irradiated area following the procedure. For transplantation, mice were placed under ketamine-xylazine anesthesia as described above. The skin area over the TA muscle was shaved and disinfected and an 11 µl cell suspension in sterile PBS + 2% FCS solution was injected into the medial portion of the TA muscle using a 25 µL, Model 702 RN SYR Hamilton Syringe coupled to a custom-made 20-mm long 26 g small hub removable needle. Mice were monitored daily. In some mice we observed skin redness and hair loss in the irradiated area at day 14–20 days post irradiation. We applied Bepanthen lotion daily on the affected skin until the date of killing. For the transplantation of PAX7null, PAX7neg, and PAX7pos cells, $10^5$ cells were injected into pre-irradiated TA muscles of NOG mice. For the transplantation of CLEC14Apos and CLEC14Aneg cells, $5 \times 10^4$ cells were injected into pre-irradiated TA muscles of NSG mice. For analysis, mice were sacrificed 21 days after cell transplantation. TA muscles were cut in two halves following a transversal plane with the cutting edge atop. Each half was separately mounted in gum tragacanth on cork disks and frozen in liquid nitrogen-cooled isopentane. Frozen muscles were stored at –80 °C.

**Intramuscular cell transplantation followed by reinjury**. In all, $5 \times 10^4$ cells were injected in pre-irradiated TA muscles of NSG mice. At day 21 post transplantation, 40 µl of 10 µM cardiotoxin (CTX, Latoxan #56574-47-1) dissolved in sterile PBS were injected in the medial portion of the TA muscle using an Omnican 50 insulin syringe (B.Braun). All the mice developed skin lesions on the irradiated skin areas starting at day 26–28 post irradiation (3–5 post-CTX) owing to a bacterial infection. They received a daily i.p. injection of enrofloxacin (BAYTRIL, 1.25 mg/ml in 0.9% NaCl solution, 240 µl/30 g body weight) and caprofen (RIMADYL, 1.25 mg/ml in 0.9% NaCl solution, 120 µl/30g body weight) from day 7–9 after CTX injection. The lesions resolved completely after the treatment and the mice were kept on Metamizole (5mg/ml in drinking water) until the end of the experiment without any further complications. Mice were sacrificed 49 days after cell transplantation (28 days after CTX injection) and TA muscles were dissected and frozen as described above.

**Electrodiagnostic (EDX) examinations**. EDX examination was performed using a Keypoint device (Natus, Middleton, WI, USA). Motor nerve conduction was measured in the median and ulnar nerves and data were interpreted referring to standard norms[52,53]. To study sensory conduction, orthodromic median mixed palmar nerve conduction was recorded and interpreted and referred to normal values. To assess neuromuscular junction function, a train of six repetitive supramaximal stimulations of the median nerve at low frequency (2 Hz) was performed.

**Sonography**. Longitudinal diaphragmatic motion was bilaterally measured using time-motion (TM), ultrasound (US) mode, and a 6 MHz transducer (Logic E ultrasound system, General Electric, Bethesda). Direct evaluation of diaphragmatic inspiratory and expiratory thickness was bilaterally obtained by brightness-modulation (B) mode measurements with a high frequency probe (12 MHz) that was positioned sagittally along the lower lateral chest wall at the so called apposition zone.

**MRI**. The proband, her parents, and an age-matched control subject underwent whole-body imaging on a free-bore 3T General Electric Discovery MR750w GEM system (General Electric, Boston, MA, USA), which employs multiple phased-array

surface receiver elements. In the whole-body configuration, subjects were wrapped in a multi-coils and multi-elements network. Head and facial muscles were explored by a head coil that is usually intended for brain imaging. Subject's safety was ensured by braces fitted to the table. The setup enabled a maximum combined field of view (FOV) of 200 cm to assure a whole-body scan. Subjects were scanned first in coronal orientation followed by axial orientation. Data acquisition was completed in five to seven steps during subject's passage from head to feed through the magnet. In the coronal orientation, the series of images that were acquired successively at an identical slice level, were automatically combined to generate a single coronal composite view using the constructor's optional software. No manual realignment was needed. The selected imaging sequences were (i) 3D T1-weighted imaging on a coronal plane with multi-stacks exploration of the body from head to mid-thigh, and (ii) IDEAL T2 derived from three points DIXON technique on axial plane with multi-stacks exploration from head to toes. Water images, fat images and in- and out-phase images were available for each slice. The recommended parameters for whole-body muscle imaging in diagnosis and out-come measures of neuromuscular disorders were respected in term of spatial resolution.

The maximal 500-mm FOV was large enough to scan all patients, including those with an important body mass. The technique required precise patient positioning with upper limbs lying close along or above the body. No paramagnetic contrast agent was injected. No cardiac or respiratory gating was performed. A multibreath-hold option was added to the thoracic sequence steps in spontaneously breathing patients who could hold their breath.

**Biplanar radiography**. Biplanar radiographs were acquired with the EOS system (Paris, France) in a standing position, using both the conventional low-dose protocol[54].

**Electron microscopy**. Samples were fixed with 2% (w/v) paraformaldehyde and 2.5% (v/v) glutaraldehyde in 0.1 M phosphate buffer. First, an initial fixation in double strength fixative and cell culture media at 37 °C in a ratio of 1:1 was done for 15 min Second, the solution was replaced with single strength fixative and incubated for 2 h at room temperature. Samples were postfixed with 1% (v/v) osmium tetroxide, dehydrated in a graded series of ethanol, and embedded in PolyBed 812 resin (Polysciences, Inc., Germany). Ultrathin longitudinal sections (60–80 nm) of the cell layer were stained with uranyl acetate and lead citrate, and examined at 80 kV with a Zeiss EM 910 electron microscope (Zeiss, Oberkochen, Germany). Acquisition was done with a Quemesa CDD camera and the iTEM software (Emsis GmbH, Germany). Images were cropped with Adobe Photoshop.

**HUVEC cell culture**. HUVECs were from PromoCell and cultured in endothelial cell growth medium 2 (PromoCell) in a humidified atmosphere containing 5% $CO_2$ at 37 °C. Medium was changed three times per week and cells were passaged at 80–90% confluence.

**Angiogenesis tube formation assay**. We performed HUVEC tube formation assay with angiogenesis μ-slides (IBIDI) as a well-established in vitro angiogenesis assays to test for endothelial characteristics in PAX7null and PAX7 proficient muscle cells. Endothelial cells are able to form capillary-like tubular structures on growth factor-reduced basement membrane extracts. We followed the application note 19 from IBIDI with some minor changes. In brief, we coated μ-slides with 10 μl growth factor-reduced Matrigel (Corning) and left it overnight covered with endothelial cell basal medium PromoCell) in a humidified atmosphere containing 5% CO2 at 37 °C. Next day, 10,000 cells per well, five replicates per cell line, were seeded in endothelial cell growth medium 2 containing 50 ng/ml FGF. Samples were imaged after 6h with an EVOS FL Cell Imaging System. Tube formation was quantified using the Angiogenesis Analyzer plugin for ImageJ.

**Statistical analysis**. All transplantation experiments were performed with four to seven biological replicates (as indicated in each legend of the figures) and results are presented as the mean. In all, 10–20 cryosections were analyzed per muscle and per mouse. For qPCR analysis, we used two reference genes and provided data against GAPDH. All statistical analysis and graphs were performed using Graph-Pad Prism Software (version 8.0). Two-tailed $t$ tests are detailed for each applicable figure.

**Reporting summary**. Further information on research design is available in the Nature Research Reporting Summary linked to this article.

## Data availability

The data sets generated during the current study are available from the corresponding author on reasonable request. Whole-exome data, bulk RNA-seq data, and single-cell sequencing data are available at. SRA accession: PRJNA481958. The SRA records will be accessible with the following link after the indicated release date: https://www.ncbi.nlm.nih.gov/sra/PRJNA481958. All source data underlying Figs. 1–4, and Supplementary Figure 1–10 are provided in Source Data file.

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

## Acknowledgements

We thank the patients without whom our research could not have been conducted. We thank Stephanie Meyer-Liesener, Stefanie Haafke, Adrienne Rothe, Carolin Gärtner, and Andrea Behm for excellent technical assistance and Susanne Wissler for help with Endnote. We also thank Denise Reichling for preparatory experiments in reconstituting *PAX7* in *PAX7*-negative muscle cells. We thank Alexej Zhogov for his help in histological characterization of transplanted cells. We thank Holger Gerhardt for having discussed lineage and experiments regarding endothial cell fate of PAX7null cells. The research was supported by the Deutsche Forschungsgemeinschaft (DFG SP 1152/12-1 to SiSp (as part of ANR-15-OE13-0011-SATNET), DFG RA 838/8-2 to NR), the International Research Training Program "MyoGrad" (DFG; IGK1631 to SiSp), the Clinician Scientist Program of the Berlin Institute of Health (J.S.), the Berlin Institute of Health (BIH CRG2aTP7 to NR), through the Helmholtz Society through Helmholtz Excellence Network for Neurocure (HFG ExNet-0036-phase2-3 to NR), the Duchenne Parents Project France, the Association Monegasque Contre les Myopathies, and the Foundation Gisela Krebs at the Max Delbrück Center for Molecular Medicine.

## Author contribution

A.M., H.E., S.A.G., E.M., J.K. designed, conducted and analyzed experiments. Sa.Sa. and S.A.G. performed and analyzed bulk-mRNA sequencing; N.K., C.K., N.R. performed and analyzed single-cell transcriptomics; A.B. performed experiments. D.P. optimized and performed irradiation of skeletal muscle. H.A. and J.S. performed clinical examination. R.C. interpreted MRI data. D.M. and S.Q.-R. performed paraclinical tests. M.S. performed whole-exome analysis. S.K. and E.M.a performed electron microscopy. A.S. provided biological specimens. A.M., H.E., S.A.G., N.K., C.B., C.K., N.R. discussed results. A.M., S.A.G., H.A., R.C. prepared figures. H.A. interpreted clinical data and wrote the clinical description of the patient. Si.Sp. designed the study, coordinated the project, raised funding, analyzed and discussed results, and wrote the manuscript.

## Competing interests

The authors declare no competing interests.
