## [Peer Review File · Nature Communications]

Reviewers' Comments:

Reviewer #1:

Remarks to the Author:

The present work by Marg et al describes the identification of human cells with myogenic potential independent of Pax7 expression. By utilizing Pax7 negative cells derived from a patient affected by Pax7 mutation in conjunction with Pax7 negative colonies derived from healthy donors, the authors report that these cells express myogenic markers desmin and Myf5 and are capable of forming myofibers upon transplantation into recipient mice with similar regenerative potential compared to Pax7-expressing clones. They further perform serial injury and demonstrate that Pax7null and Pax7neg colonies exhibit self-renewal ability upon transplantation into recipient mice. By single cell gene expression profiling the authors observed significant heterogeneity within and among the different samples. In the colonies derived from the patient affected by a Pax7 mutation, the authors observed elevated expression of endothelial markers TFPI2 and CLEC14A. Finally, WT Pax7 overexpression in Pax7 mutant cells recapitulated myogenic gene expression and cell morphology. The authors conclude that myogenic potential and regenerative capacity is independent of Pax7 expression and that this work identifies a novel myogenic population. The manuscript is well written, however, additional evidence should be provided in order to fully support the claim that CLEC14A marks a novel myogenic population independent of Pax7.

Main points:

1- The claim that CLEC14A+ cells are a novel myogenic population independent of Pax7 is not fully supported by the results shown. In Figure 1 the authors show that Pax7null cells from the affected patient have elevated levels of Pax3 expression as well as PDGFRalpha and OSR1, marker of fibroadipogenic progenitors (FAP). The authors conclude that "Pax7null colonies did not fall in one of the previously described categories of interstitial muscle cells.". It is unclear how they make this conclusion. In single cell RNAseq shown in Figure 2 the authors show that Pax7null and Pax7neg colonies are quite different in composition and gene expression, and Pax7null cells express CLEC14A, Pax3, Myf5 as well as FAP genes PDGFRalpha and OSR1. Thus, it cannot be ruled out that they could be a subpopulation of FAP or Pax3 myogenic cells. The authors should tone down their conclusions, and discuss this possibility, or provide lineage tracing evidence in mouse models to conclusively distinguish between these two possibilities.

2- In Figure 3 and 4 the authors show expression of CLEC14A underneath the basal lamina in transplantation assays and in the affected patient muscle. However, quantification should be shown to evaluate the frequency of these events.

3- Human cell colonies are selected in culture, and although the authors provide an image of CLEC14A positive cell in human tissue sections, there is no quantification and it is unclear what is the abundance of this cell population in tissues or upon fresh isolation of cells from tissues, in the absence of a culture selection step. This issue should be addressed to avoid misinterpreting the findings derived from culture-expanded cells. The sentence "CLEC14A positive cells belong to the normal spectrum in humans and are not confined to the rare patient carrying the Pax7 null mutation" is not fully supported. In Figure 4e, what is the abundance of CLEC14A positive nuclei in the affected patient tissue, and how does it compare with healthy patient muscle tissues?

4- A recently published manuscript (Feichtinger et al, Genetics in Medicine, May 2019), describes the skeletal muscle phenotype of five patients affected by mutations in the Pax7 gene. The study

reports patients myopathy. They further report the presence of a small population of desmin positive myogenic cells in the absence of Pax7. This work should be cited, and the current findings reconciled with this previous work.

5- Statistical analysis, sample size, how many times was the experiment repeated, statistical method used, and P value is missing in all Figure legends.

Specific Points:

1- In Figure 3a, the control of Pax7pos cells should be shown in the fusion index assessment.

2- For Figure 4d, a representative FACS plot of CLEC14A sorting should be shown.

Reviewer #2:

Remarks to the Author:

The authors have made extensive revisions to the original manuscript and have greatly strengthened the paper. A few questions remain:

1. Can you include more FAP/pericyte/fibroblast/endothelial/twist markers in the analysis. It is still not how to call/describe these cells. Are they all homogenous or heterogenous with regard to these markers within individual cells?

2. It seems reinjury has been performed but there is no quantification of the data or clear description of the findings. Is there more regeneration/more/larger myofibers, or other signs of repopulation after reinjury?

3. Clearly these cells can not make up for the loss of PAX7 again suggesting they are not truly capable of regenerating muscle back to wild type levels. This should be considered/discussed especially with regard to the conclusions the authors are making that this novel population should be considered as an alternative/better population for cell-based therapies as this may not be the case.

4. Have you performed overlap gene set analysis with recent PAX3 reserve cell papers from Brack and Relaix? Are these similar? Do they have similar functional responses as seen in these mouse studies?

5. Have you evaluated overexpression of CLEC14A in PAX7 null cells in vitro? Plus and minus this OE could help determine more functional potential/identification of this population.

Reviewer #3:

Remarks to the Author:

The Authors have satisfactorily replied to all the concerns raised. I have no further comments.

Answers to the reviewers

We would like the reviewers for their time and input.

Reviewer 1

The claim that CLEC14A+ cells are a novel myogenic population independent of Pax7 is not fully supported by the results shown. In Figure 1 the authors show that Pax7null cells from the affected patient have elevated levels of Pax3 expression as well as PDGFRalpha and OSR1, marker of fibroadipogenic progenitors (FAP). The authors conclude that "Pax7null colonies did not fall in one of the previously described categories of interstitial muscle cells.". It is unclear how they make this conclusion. In single cell RNAseq shown in Figure 2 the authors show that Pax7null and Pax7neg colonies are quite different in composition and gene expression, and Pax7null cells express CLEC14A, Pax3, Myf5 as well as FAP genes PDGFRalpha and OSR1. Thus, it cannot be ruled out that they could be a subpopulation of FAP or Pax3 myogenic cells. The authors should tone down their conclusions, and discuss this possibility, or provide lineage tracing evidence in mouse models to conclusively distinguish between these two possibilities.

Our answer: All RNA-Seq and single cell sequencing data from our human muscle stem cell populations are available as open source information of our manuscript in a convenient online tool (https://shiny.mdc-berlin.de/hummus_sc_XkZL9gHZE2UBwjGb/). Therefore, all molecular properties of our newly described muscle stem cell population are available. We agree with the reviewer's suggestion to state our conclusions more cautiously. We changed our statement to "CLEC14A positive myogenic cells display features of FAPs because they express *PDGFRa* and *OSR1*. Also, they do not express PW1 which distinguishes them from PICs. They also do not share the features of mesangioblasts (CD133). Furthermore, in contrast to FAPs or mesangioblasts, CLEC14A positive myogenic cells inhabit the satellite cell niche. Apart from CLEC14A, other endothelial cell markers such as ERG1, PECAM, or VCAM were not expressed in CLEC14A positive human myogenic cells. We conclude that there is an overlap between features of CLEC14A positive myogenic cells and previously described cell populations, but CLEC14A appears to define a so far non-described population in human muscle." (Manuscript page 8, paragraph 2).

2- In Figure 3 and 4 the authors show expression of CLEC14A underneath the basal lamina in transplantation assays and in the affected patient muscle. However, quantification should be shown to evaluate the frequency of these events.

Our answer: Quantification of CLEC14A cells is now provided in Figure legends Fig. 4d and Fig. 4e. Raw data are demonstrated in the Source data sheets. CLEC14A positive cells in satellite cell position in the PAX7null patient appear in approximately the same frequency (8%) as PAX7 positive cells in healthy individuals during childhood.

3- Human cell colonies are selected in culture, and although the authors provide an image of CLEC14A positive cell in human tissue sections, there is no quantification and it is unclear what is the abundance of this cell population in tissues or upon fresh isolation of cells from tissues, in the absence of a culture selection step. This issue should be addressed to avoid misinterpreting the findings derived from culture-expanded cells. The sentence "CLEC14A positive cells belong to the normal spectrum in humans and are not confined to the rare patient carrying the Pax7 null mutation" is not fully supported. In Figure 4e, what is the abundance of CLEC14A positive nuclei in the affected patient tissue, and how does it compare with healthy patient muscle tissues?

Our answer: In addition to quantification of CLEC14positive cells in sections of the PAX7null patients we also attempted to identify CLEC14A positive cells in satellite cell position in frozen muscle sections from healthy individuals. This proved to be difficult. As we show in Fig. 4b, expression of PAX7 and CLEC14 is mutually exclusive in fiber-derived cell populations. This suggests that CLEC14A is down-regulated in the presence of PAX7, a notion also supported by Figure 4c. In addition, bright positivity for CLEC14A-positive capillary-endothelial cells in skeletal muscle renders interpretation of immunofluorescent stainings difficult. The figure below provides an example of CLEC14A immunofluorescent stain of normal human frozen muscle sections.

We therefore employed smFISH techniques to identify *CLEC14A*-mRNA-positive cells in satellite cell position and indeed, as a rare event, we can demonstrate mRNA molecules clearly distinct from endothelial cell position. This is included now in Fig. 4 (Fig. 4d). Appropriate methodological information is provided in Material and Methods (page 11).

Fig. 1: CLEC14A (red) staining of normal skeletal muscle in frozen sections (10 μ m). Left: anti-CLEC14A, red; right: Hoechst. Bar, 50 μ m.

In summary, the presence of CLEC14A positive cell populations derived from 15-20% of human muscle fiber fragments, and their depiction in satellite cell position in the PAX7null patient plus in healthy individuals demonstrates that CLEC14A cells are not only confined to the rare pathological variant of lacking PAX7.

4- A recently published manuscript (Feichtinger et al, Genetics in Medicine, May 2019), describes the skeletal muscle phenotype of five patients affected by mutations in the Pax7 gene. The study reports patients myopathy. They further report the presence of a small population of desmin positive myogenic cells in the absence of Pax7. This work should be cited, and the current findings reconciliated with this previous work.

Our answer: The publication by Feichtinger et al. is now included into the references (Ref. Nr 17, page 3). The finding of a desmin-positive, PAX7-negative population is in concordance with our manuscript.

5- Statistical analysis, sample size, how many times was the experiment repeated, statistical method used, and P value is missing in all Figure legends.

Our answer: Careful quantitative analysis has always been performed in all experiments. Differences between newly generated fibers after transplantation and after re-injury between PAX7pos, PAX7neg, and PAX7null transplants could not be observed. A “Not-significant” indication and the statistical method have now been included in the legends of Figures 3 and 4 and Supplementary Fig. 3. Each dot in the quantification graph has represented one transplanted mouse. Supplementary table 2 indicates which cell population has been used for which experiment. Raw data are available as Source data file. Sample size is also included in Material and Methods.

Specific Points:

1- In Figure 3a, the control of Pax7pos cells should be shown in the fusion index assessment.

Our answer: This has now been included into Fig. 3a.

2- For Figure 4d, a representative FACS plot of CLEC14A sorting should be shown.

The FACS sorting strategy is now depicted as Supplementary Fig. 8.

Reviewer #2 (Remarks to the Author):

The authors have made extensive revisions to the original manuscript and have greatly strengthened the paper. A few questions remain:

1. Can you include more FAP/pericyte/fibroblast/endothelial/twist markers in the analysis. It is still not how to call/describe these cells. Are they all homogenous or heterogenous with regard to these markers within individual cells?

Our answer:

A comprehensive marker analysis is provided. The following table summarizes the markers, the relevant literature and the position in the manuscript in which the respective marker analysis is shown. The figure below also gives an example of how the comfortable online tool of the single cell sequencing data can be used to check for any marker that a potential reader may want to search for. https://shiny.mdc-berlin.de/hummus_sc_XkZL9gHZE2UBwjGb/. A more detailed discussion is included in the manuscript (page 6).

Summary: Lineage studies in Marg et al., NCOMMS-19-14307-T

Lineage defining genes	Lineage	Expression in PAX7null compared to PAX7pos cells	Shown in	Key reference	Cited in paper ¹
PAX7	Muscle stem cell	Not detected	Figs. 1-4, Suppl. Fig. 1-7, 10	Seale et al., Cell , 2000 and more	yes
MYF5	Muscle stem cells, myogenic lineage	Not different	Figs. 1c,d,2,4 Suppl. Fig. 4-7	Beauchamp JR, et al., J Cell Biol , 2000	yes
PAX3	Myogenic lineage	Higher	Fig. 1c, 2, Suppl. Fig. 4-7	Relaix F, et al., Nature 2005	yes
MYOD1	Myogenic lineage	Lower	Fig. 1c, 2, 4c, Suppl.Figs.4-7		
NCAM1	Surface marker myoblasts, muscle stem cells, endothelial cells	Lower	Fig. 1c,d, Fig.2,4, Suppl. Fig. 2, 4-7	Illa I, et al., Ann Neurol 1992	no
PDGFRa	Fibroadipogenic cells (FAPs)	Higher	Fig. 1c, Fig. 2, Suppl.Fig. 4-7	Uezumi et al Nat Cell Biol 2010	yes
PW1/PEG3	PW1 interstitial cells (PICs)/FAPs	Lower	Fig. 1c, Fig. 2, Suppl Fig. 4-7	Mitchell KJ et al., Nat Cell Biol 2010	yes
DES	Myogenic lineage	Not different	Fig. 1,2,4, Suppl. Fig. 3-7, 10		
OSR1	Activated FAPs	Higher	Fig. 2, Suppl. Fig. 2, 4-7	Vallecillo-Garcia P, et al., Nat Commun , 2017	yes
ITGB1/CD29	FAPs, Mesenchymal stem cells, many cell types	Not different	Fig. 2, Suppl. Fig. 2, 4-7	Péault B, et al., Mol Ther , 2007	yes
TCF4	Muscle connective tissue	Not different	Fig. 2, Suppl. Fig. 2, 4-7	Kardon G, et al., Dev Cell , 2003	yes
TP53	PICs	Not different	Fig. 2, Suppl. Fig. 2, 4-7	Mitchell KJ et al., Nat Cell Biol 2010	yes
CD34	FAPs, PICs in mice,	Not detected	Human cells do not express the	Joe AW et al., Nat Cell Biol , 2010	no

			marker in myogenic cells		
SCA1 ²	FAPs	Not different	Fig. 2, online tool	Mitchell KJ et al., Nat Cell Biol 2010	yes
CD133/ PROM1	Mesangioblasts	Not detected	Fig. 2, Suppl. Fig. 2, online tool	Minasi MG et al., Development , 2002	yes
ACTN1	Muscle	Not different			
TFPI2	Blood vessels, esophageal cancer, cardiogenesis	Higher	Fig. 2, Suppl. Fig. 4-7	Crawley, JT et al. Art Thromb Vasc Biol. , 2002	yes
CLDN11	Tight junctions, endothelial cells	Higher	Fig. 2, Suppl. Fig. 4-7	Gow, A et al., Cell , 1999 Li, B et al., Endocr Res , 2017	yes
FHL1	Muscle, heart	Higher	Fig. 2, Suppl. Fig. 2, 4-7	Schessi J et al., J Clin Inv , 2008 Knoblauch H et al., Ann Neurol , 2010	yes
COL6A3	Extracellular matrix	Higher	Fig. 2, Suppl. Fig. 2, 4-7	Camacho Vanegas et al. PNAS , 2001	yes
ERG 1	Endothelial cells	Not different	Fig. 2, Suppl. Fig.4-7,10	Birdsey GM, et al., Blood , 2008	yes
VCAM	Endothelial cells	Not different	Fig. 2, Suppl. Fig.4-7,10		no
PECAM	Endothelial cells	Not different	Fig. 2, Suppl. Fig.4-7,10	Dasgupta B, et al., J Immunol , 2009	yes
CLEC14A	Endothelial cells	High	Topic of manuscript		

Fig.: Example of usage on single cell sequencing online tool https://shiny.mdc-berlin.de/hummus_sc_XkZL9gHZE2UBwjGb/. Genes of interested can easily be visualized. Additional data are provided in the data sets of RNASeq bulk sequencing. Here: SCA1 gene expression is low in PAX7null, PAX7neg, and PAX7 high cell population.

2. It seems reinjury has been performed but there is no quantification of the data or clear description of the findings. Is there more regeneration/more/larger myofibers, or other signs of repopulation after reinjury?

Our answer: Careful quantitative analysis has always been performed in all experiments. Differences between newly generated fibers after transplantation and after re-injury between PAX7pos, PAX7neg, and PAX7null transplants could not be observed. This holds true for the absolute number of newly generated muscle fibers as well as for the morphology of individual fibers as demonstrated in Figures 3b, Figure 4d, and Supplementary Fig. 3. Bar graphs are included in all figures. A “Not-significant” indication and the statistical method have now been included in the legends of Figures 3 and 4 and Supplementary Fig. 3. Each dot in the quantification graph has represented one transplanted mouse. Supplementary Table 2 indicates which cell population has been used for which experiment. Raw data are available as Source data file. Sample size is also included in Material and Methods.

3. Clearly these cells can not make up for the loss of PAX7 again suggesting they are not truly capable of regenerating muscle back to wild type levels. This should be considered/discussed especially with regard to the conclusions the authors are making that this novel population should be considered as an alternative/better population for cell-based therapies as this may not be the case.

Our answer: We surely did not want to suggest that CLEC14A-positive cells are a *better* population for cell-based therapies. We changed our wording to more careful statements and conclude our discussion with “They have myogenic and regenerative capacity and could become relevant in cell therapies involving muscle stem cells.”

4. Have you performed overlap gene set analysis with recent PAX3 reserve cell papers from Brack and Relaix? Are these similar? Do they have similar functional responses as seen in these mouse studies?

Our answer: The studies from the Brack and Relaix group, both Cell Stem Cell, June 2019, are now cited in the paper. These studies are very interesting as they provide evidence that the satellite cell population is heterogeneous and that PAX3 positive cells are better equipped to fight metabolic/ environmental stress. Indeed, we find that PAX3 is expressed in muscle stem cells derived from the PAX7null patient. Extensive gene set analyses of the reserve cell populations is not provided in the manuscripts. A comparison would be interesting.

5. Have you evaluated overexpression of CLEC14A in PAX7 null cells in vitro? Plus and minus this OE could help determine more functional potential/identification of this population.

CLEC14A is approximately 40fold overexpressed in PAX7null cells as demonstrated in Figure 4c. All gene expression data resulting from this overexpression are available in Figure 2, Figure 4, Supplementary Fig. 2, 4-7.

Reviewer #3 (Remarks to the Author):

The Authors have satisfactorily replied to all the concerns raised. I have no further comments.

Reviewers' Comments:

Reviewer #2:

Remarks to the Author:

Although some of the claims are still strong and the population not completely clear the authors have attempted to respond to reviewer comments.

Reviewer #4:

Remarks to the Author:

Marg and colleagues' studied in vitro myogenesis from human muscle samples. They identified a myogenic cell population from Pax7-mutated patient sample. They further conduct in-depth molecular and cellular profiling of these Pax7-null cells compared with healthy myoblasts colonies expressing Pax7 or not (Pax7-pos, Pax7-neg).

This review will only focus on the single-cell transcriptomics aspects.

scRNA-seq was performed using the Drop-Seq technology. High numbers of cells were processed, with a good number of transcripts per cell (> 2,000 UMI and scRNA-seq usually exclude cells with fewer than 1,000 UMI). The authors used FastQC and DigitalExpressionTool for raw data analysis. They then used the R packages Dropbead (tailored for Drop-Seq) and Seurat (a very reliable tool) for dimensionality reduction and representation of gene expression. All tools and techniques are state-of-the-art. The analysis is sound and valid.

However, I have two concerns:

1. The studied population are very closely related, and the t-SNE maps do not readily allow for evaluation of the differences. The authors should provide the readers with a list of the most discriminating genes between the three samples; Pax7-pos/-neg/-null.

2. The authors state "We found highly expressed genes with no previous association to myogenic progenitor cells, such as TFPI2, CLDN11, CLEC14A, COL6A3, and FHL1."

This is not true, since:

a. FHL1 is expressed by myogenic cells, and mutations in this gene are causal to Emery-Dreifuss Muscular Dystrophy (PMID: 19716112 ; PMID: 19075112).

b. TFPI2 has been shown to be expressed by activated satellite cells (PMID: 19962952).

Response to the Reviewers

We thank the reviewers for their time and effort as they helped us to improve the manuscript.

Reviewer #2 (Remarks to the Author):

Although some of the claims are still strong and the population not completely clear the authors have attempted to respond to reviewer comments.

We appreciate the remark.

Reviewer #4 (Remarks to the Author):

Marg and colleagues' studied in vitro myogenesis from human muscle samples. They identified a myogenic cell population from Pax7-mutated patient sample. They further conduct in-depth molecular and cellular profiling of these Pax7-null cells compared with healthy myoblasts colonies expressing Pax7 or not (Pax7-pos, Pax7-neg).

This review will only focus on the single-cell transcriptomics aspects.

scRNA-seq was performed using the Drop-Seq technology. High numbers of cells were processed, with a good number of transcripts per cell (> 2,000 UMI and scRNA-seq usually exclude cells with fewer than 1,000 UMI). The authors used FastQC and DigitalExpressionTool for raw data analysis. They then used the R packages Dropbead (tailored for Drop-Seq) and Seurat (a very reliable tool) for dimensionality reduction and representation of gene expression. All tools and techniques are state-of-the-art. The analysis is sound and valid.

However, I have two concerns:

1. The studied population are very closely related, and the t-SNE maps do not readily allow for evaluation of the differences. The authors should provide the readers with a list of the most discriminating genes between the three samples; Pax7-pos/-neg/-null.

A full table listing discriminating genes and quantification measures between the populations is provided as Supplementary table 9 in a separate Excel file. The reader and the reviewers are also encouraged to use the comfortable online tool that is designed to visualize all genes of interests in the clusters as well as providing lists with most highly expressed genes in each cluster and comparing myogenic populations.

<https://shiny.mdc-berlin.de/humuscl/>

2. The authors state "We found highly expressed genes with no previous association to myogenic progenitor cells, such as TFPI2, CLDN11, CLEC14A, COL6A3, and FHL1."

This is not true, since:

a. FHL1 is expressed by myogenic cells, and mutations in this gene are causal to Emery-Dreifuss Muscular Dystrophy (PMID: 19716112 ; PMID: 19075112).

b. TFPI2 has been shown to be expressed by activated satellite cells (PMID: 19962952).

We agree with the reviewer.

- Indeed, *Tfpi2* is mentioned in a list (Figure 6C) of the cited reference on mouse satellite cells by Pallafacchina et al., 2010. Cultured mouse myoblasts failed to express *Tfpi2*.
- In zebrafish-knockout for *fhl1*, the fish had developmental abnormalities and a reduced number of satellite cells (PMID29521230).

Both references have now been included into the cited literature. We changed the wording of the sentences to:

“We found a number of genes highly expressed in the PAX7null population only, that so far mainly had been studied in a context other than satellite cell biology, *TFPI2*, *CLDN11*, *CLEC14A*, *COL6A3*, and *FHL1*”